# Effects of different roughness on Brazilian splitting characteristics of rock-concrete interface

Yan Chen[1,2,3,4]*, Gaofei Wang[1,4], Lei Zhou[1,4], Liangtao Deng[1,4], Jiahao Wang[1,4]

**1** School of Energy Science and Engineering, Henan Polytechnic University, Jiaozuo, China, **2** Henan Key Laboratory for Green and Efficient Mining & Comprehensive Utilization of Mineral Resources, Henan Polytechnic University, Jiaozuo, China, **3** Collaborative Innovation Center of Coal Work Safety and Clean High Efficiency Utilization, Henan Polytechnic University, Jiaozuo, China, **4** Jiaozuo Road Traffic and Transportation Science and Technology Research Center, Henan Polytechnic University, Jiaozuo, China

* chenyan@hpu.edu.cn

**Data Availability Statement:** All relevant data are within the manuscript and its Supporting Information files.

**Funding:** This study was supported by the National Natural Science Foundation of China (51904092),

## Abstract

In order to study the tensile properties of rock-concrete composite disc specimens with different roughness, the surface of the gray-white sand specimen was artificially grooved, and six different roughness were configured. The test results show that the roughness size and roughness mode jointly control the tensile strength of the rock-concrete interface. With the increase of roughness, the tensile strength of the sample changes from the initial decrease to the increase and then decrease, and the tensile strength reaches the highest when the roughness is $f_3$. The variation trend of pre-peak energy accumulation and post-peak energy accumulation of the sample is opposite, and the dissipation energy is closely related to the crack propagation strain. The roughness and crack closure strain, crack peak strain, crack propagation strain and crack closure stress show a sinusoidal periodic variation. The crack propagation strain is closely related to the change of dissipation energy. The change trend of crack closure stress is basically consistent with the change trend of tensile strength. Therefore, in the actual project, grasping the period of roughness variation and selecting the construction position can make the rock-concrete interface stable and get twice the result with half the effort.

## 1. Introduction

Due to the good bonding performance between rock and concrete, concrete lining structures are widely used in tunnel engineering and mine systems [1–3]. However, the interface transition zone (ITZ) will be formed between rock and concrete. Under the influence of the ' wall effect ' of the rock interface, the porosity in the ITZ is high and the bonding strength is weak. The interface is considered to be the weak surface of the rock-concrete composite structure. Structural failure often occurs on these weak surfaces [4–7]. In order to ensure structural safety and eliminate engineering hidden dangers, it is urgent to study the mechanical properties of the rock-concrete interface.

Henan Science and Technology Research Project (242102321163), Key Scientific Research Project of Higher Education Institutions in Henan Province (24A440004), the Fundamental Research Funds for the Universities of Henan Province (NSFRF230403, NSFRF210454), Young backbone teachers funding program of Henan Polytechnic University (2022XQG-01), the research fund of Henan Key Laboratory for Green and Efficient Mining & Comprehensive Utilization of Mineral Resources (KCF2202) and the research fund of Jiaozuo Road Traffic and Transportation Science and Technology research center (JRTT2023004, JRTT2023010, JRTT2023011). The funders had no role in study design, data collection and analysis, decision to publish, or preparation of the manuscript.

**Competing interests:** The authors have declared that no competing interests exist.

The mechanical properties of rock-concrete interface are greatly related to the material properties of rock and concrete [8–10], interface roughness [11–13], concrete pouring technology [14] and so on. In recent years, the research methods on the mechanical properties of rock-concrete interface are mainly through compression test [15–18], shear test and tensile test. [15] discussed the bonding behavior of the interface between rock and mortar, and pointed out that the interfacial bonding force increased with the increase of surface roughness, but the increase gradually decreased. [17] obtained by triaxial compression test that the axial peak stress and axial peak strain of the specimen are significantly higher than those of the specimen without contact surface constraint when the rock-concrete contact surface is constrained [18]. The rock-like-concrete composite under smooth interface and wavy interface is studied by compression test. It is found that with the increase of roughness, most of the performance parameters will be significantly reduced, and only a few performance parameters will be improved. The static and dynamic damage degree of the composite under the wavy interface is more serious than that under the smooth interface. Through the shear test of sandstone-concrete specimens with natural rough surface, the shear activity of sandstone-concrete interface was analyzed. The shear behavior of rock-concrete under constant normal stiffness is studied by using triangular roughness instead of natural rough surface [19]. The indoor direct shear test results of standard cylindrical specimens with different surface roughness of rock were carried out. It was found that with the increase of roughness, the brittleness and peak dilatancy of the specimens increased, while the residual shear strength showed the opposite trend. The effect of groove depth on the mechanical properties of rock-concrete interface was studied. Their research results show that with the increase of groove depth, the bonding performance of rock-concrete interface increases first and then tends to be stable. studied the effect of serrated interfaces with different heights on the tensile strength of the rock-concrete interface, and pointed out that the interface roughness determines the bearing capacity of the specimen by affecting the mechanical interaction of the interface. The influence of serrated width of rock surface on the dynamic performance of rock-concrete interface is studied. It is proposed that there is an optimal groove width at the interface, which can improve the tensile and shear strength of the interface.

In summary, Brazilian splitting test has made relatively mature progress in rock single splitting under different influencing factors. It provides certain theoretical research value for tunnel engineering in terms of tunnel failure mode, failure characteristics and fracture strength characteristics. However, there are still few studies on the relationship between the influencing factors of the interface roughness of rock-concrete composite specimens and the failure mode of rock mass.

Undoubtedly, the study of rock-concrete interface failure by Brazilian splitting test method will enrich the current understanding of the tensile fracture characteristics of rock-concrete interface. Therefore, based on the current research results, in order to better understand the tensile instability failure mechanism of rock-concrete interface, this study explores the influence of interface roughness on the tensile strength, deformation, splitting energy rate and failure mode of rock-concrete composites. In this study, Brazilian splitting tests were carried out on six disc specimens with different roughness by grooving to quantify the roughness.

## 2. Test overview

### 2.1 Sample preparation

In this work, the rock sample is gray-white sand, which is taken from Jiaozuo stone factory. It is gray-white and produced in Zigong, Sichuan Province. The water absorption rate is 3.33%. The concrete samples used in the test are made of cement, water, fine aggregate and coarse

**Table 1. Concrete mix ratio.**

| Materials | Cement | Water | Medium Sand | Gravel |
|---|---|---|---|---|
| Proportion (%) | 45.36 | 10.71 | 8.12 | 35.81 |

aggregate as raw materials, and mixed evenly according to the appropriate proportion. Among them, the fine aggregate adopts medium sand, the coarse aggregate adopts the maximum particle size of 5 mm mechanism gravel, the cement adopts ordinary Portland cement, and the aggregate gradation adopts continuous gradation, as shown in Table 1.

The mold adopts a PVC heavy-duty circular tube with an inner diameter of 50 mm and a height of 30 mm. This material has high stability and is not suitable for compression deformation. Before preparing the sample, the inner wall of the tube is evenly coated with a layer of concrete water-based release agent film, and the antistaling film is used to ensure that there is no liquid leakage.

## 2.2 Sample making

(1) **Treatment of rock-concrete interface with different roughness.** The processing flow and equipartition groove of sandstone are shown in Fig 1. Firstly, a standard cylindrical sandstone sample with a diameter of 50 mm and a thickness of 30 mm was vertically cut into a semi-cylinder along the diameter direction using a diamond saw blade with a diameter of 150 mm and a thickness of 2 mm. Six grooves with different roughness were machined on the cross section of the cuboid, denoted as $f_i$ ($i$ = 0,1,2,3,4,5), respectively. Its number corresponds to the number of slots. The diameter is divided into $i$+1 equal parts along the diameter direction. The equipartition point is the midpoint of the side length of the notch area in the diameter direction of the sample. Each notch area is a cuboid perpendicular to the bottom of the sample, with a size of 3 mm ×3 mm ×30 mm. Three samples are processed in each group, a total of 18 samples.

(2) **Splitting principle Interface roughness measurement.** Sand filling method is a simple, convenient and practical method for measuring the roughness of concave-convex uniform interface. After the grooving treatment of the rock surface, the average depth of the sand filling

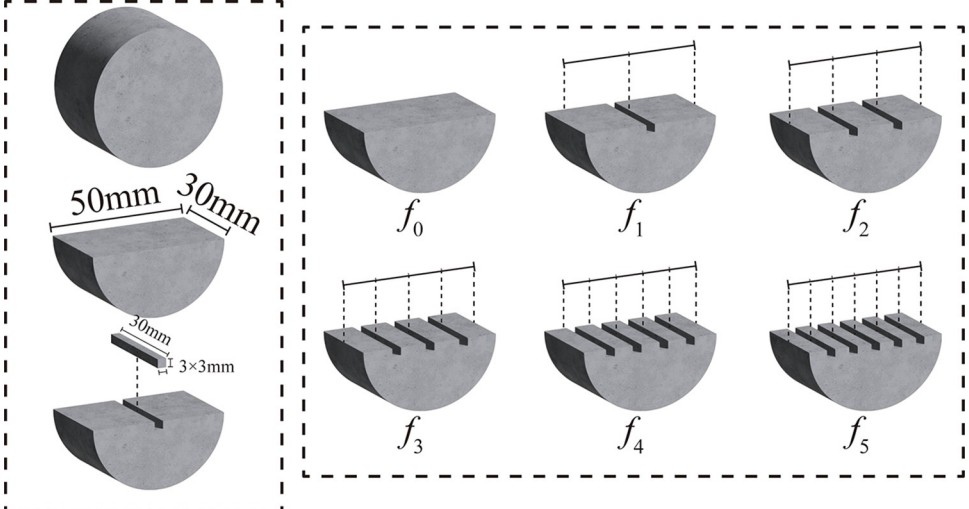

**Fig 1. Schematic diagram of sandstone processing flow and equal division groove.**

method is used to characterize the roughness of the rock-concrete interface to quantify its effect on the tensile strength. The specific operation is as follows:

The rock to be measured is placed on a horizontal table. In order to prevent the spilled sand from leaking, the rock is surrounded by a plastic plate in advance, and the top of the plastic plate and the highest point of the rock surface are flushed. Then the fine sand is evenly spread on the rough surface of the rock and filled with grooves. In order to make the fine sand not exceed the initial surface of the rock, the surface needs to be flattened. The volume of fine sand spread on the rock surface was measured three times, and the measurement results were recorded. The average depth of the sand filling method was obtained by dividing the sand filling volume by the rock surface area, which was used to characterize the roughness of the rock-concrete interface [20, 21].

$$f_i = \frac{V}{A} \tag{1}$$

Where $f_i$ is the interface roughness ($i$ = 0, 1, 2, 3, 4, 5); V is sand filling volume (mm$^3$); and A is the interface cross-sectional area (mm$^2$).

The calculated values of different roughness are shown in Table 2.

With the increase of the number of grooves, the contact area between rock and concrete becomes larger, and the interface roughness increases.

**(3) Making rock-concrete standard disc specimen.** The arc surface of the rock sample after the groove is slowly placed along the inner wall of the PVC pipe, and the mixed concrete is poured into the empty space of the mold. After the vibration, it is ensured that the arc

**Table 2. Roughness calculation.**

| Roughness | NO. | D/mm | H/mm | V/mm$^3$ | $f_i$/mm |
|---|---|---|---|---|---|
| $f_0$ | $f_0$-1 | 49.13 | 29.98 | 0 | 0 |
| | $f_0$-2 | 49.02 | 30.40 | 0 | 0 |
| | $f_0$-3 | 48.25 | 30.06 | 0 | 0 |
| | Mean value | 48.80 | 30.15 | 0 | 0 |
| $f_1$ | $f_1$-1 | 48.24 | 30.03 | 1448.65 | 0.21 |
| | $f_1$-2 | 48.25 | 30.90 | 1490.93 | 0.21 |
| | $f_1$-3 | 49.26 | 31.90 | 1571.40 | 0.19 |
| | Mean value | 48.58 | 30.94 | 1503.07 | 0.20 |
| $f_2$ | $f_2$-1 | 49.04 | 30.42 | 1491.80 | 0.47 |
| | $f_2$-2 | 48.51 | 30.35 | 1472.28 | 0.51 |
| | $f_2$-3 | 49.09 | 30.48 | 1496.26 | 0.48 |
| | Mean value | 48.88 | 30.42 | 1486.93 | 0.49 |
| $f_3$ | $f_3$-1 | 48.82 | 30.24 | 1476.32 | 0.68 |
| | $f_3$-2 | 49.36 | 30.46 | 1503.51 | 0.63 |
| | $f_3$-3 | 48.85 | 30.57 | 1493.35 | 0.64 |
| | Mean value | 49.01 | 30.42 | 1490.89 | 0.65 |
| $f_4$ | $f_4$-1 | 49.85 | 30.84 | 1537.37 | 0.94 |
| | $f_4$-2 | 49.19 | 30.44 | 1497.34 | 0.93 |
| | $f_4$-3 | 48.86 | 30.34 | 1482.41 | 0.95 |
| | Mean value | 49.30 | 30.54 | 1505.62 | 0.94 |
| $f_5$ | $f_5$-1 | 49.14 | 30.31 | 1489.43 | 1.15 |
| | $f_5$-2 | 48.77 | 31.05 | 1514.31 | 1.16 |
| | $f_5$-3 | 48.54 | 30.51 | 1480.96 | 1.15 |
| | Mean value | 48.82 | 30.62 | 1494.87 | 1.15 |

surface of the rock sample is closely bonded with the PVC pipe and then wrapped with the antistaling film, which is used to prevent the evaporation of the water in the concrete. Then, it is placed in the room with a temperature of 25°C and a relative humidity of 95% for 24 hours, and the mold is removed and placed in the standard curing room for curing. The rock-concrete composite specimen was taken out, and the size error of the specimen was controlled within ± 0.5 mm after grinding and polishing. The six roughness rock-concrete specimens are shown in Fig 2.

Fig 2(A) shows the positional relationship between sandstone and concrete. Due to the characteristics of the processing method, in Fig 2(B), it can be seen that the enlarged interface groove is like the shape of the letter U, and the processing error of all grooves can be ignored by the unified method.

The Brazilian splitting test adopts the RMT-150 B digitally controlled electro-hydraulic servo testing machine. The diameter of the steel wire spacer is 2 mm and is pressed at the interface of the rock-concrete interface of the sample. The test instrument and the sample placement method are shown in Fig 2(C). The force sensor of this test: vertical upward, and the range of the force sensor perpendicular to the force of the test piece is 1000 KN; the position sensor of the vertical deformation of the test piece is 5 mm in the vertical direction, and two sensors are 2.5 mm in the horizontal direction (transverse deformation of the test piece). The measurement process is displayed on the digital screen of the computer, and the vertical stress, vertical strain and lateral deformation are automatically measured by the system.

It is not difficult to find that when the number of grooves is odd and even, the center of the sample is divided into two cases: the center with concrete serrations and the center without

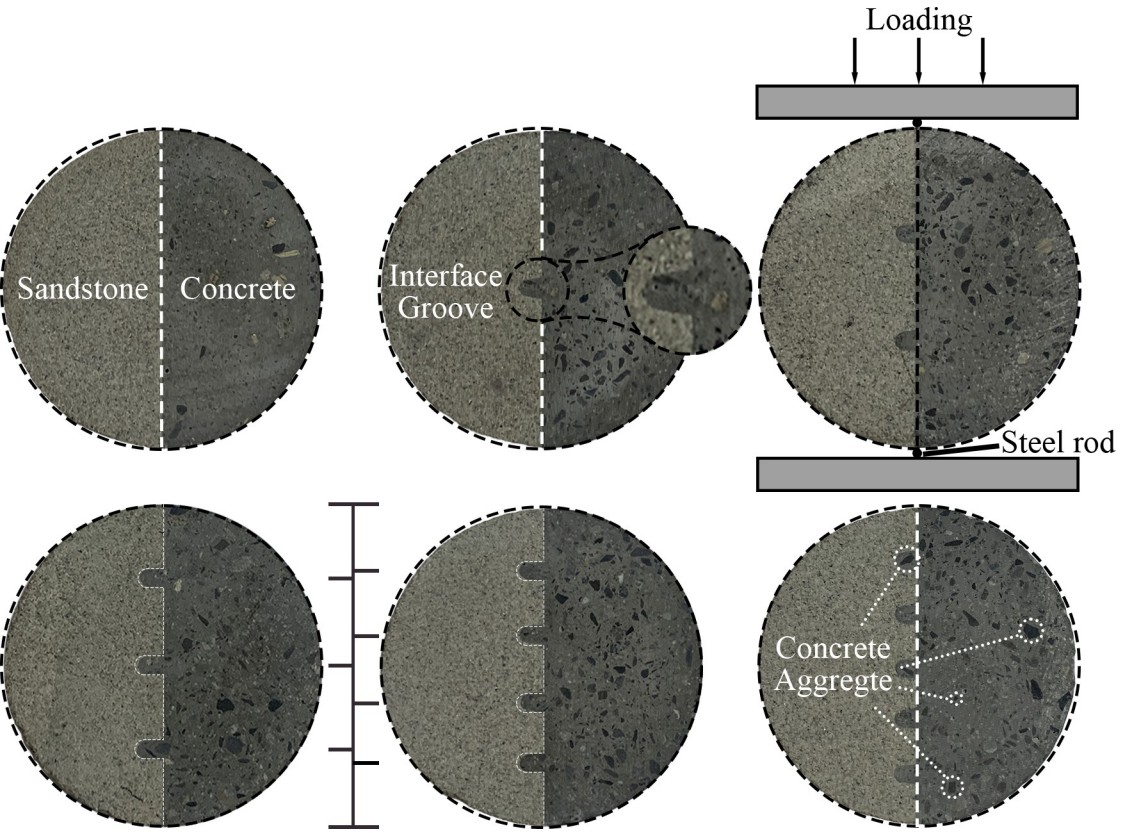

**Fig 2.** Sample groove type and loading diagram, (a) $f_0 = 0$; (b) $f_1 = 0.20$; (c) $f_2 = 0.49$; (d) $f_3 = 0.65$; (e) $f_4 = 0.94$; (f) $f_5 = 1.15$.

serrations. As shown in Fig 2(D) and 2(E), with the help of the auxiliary line between the two figures of Fig 2(D) and 2(E), it can be seen that the diameter is divided into 4 parts, and the number of grooves is odd. At this time, the middle part of the sample diameter has concrete serrations, and the diameter is divided into 5 parts. The number of grooves is even, and there is no concrete serration in the middle of the sample diameter. The distribution of aggregates with different particle sizes can be clearly observed on the surface of concrete, and some annotations are made in Fig 2(F).

## 3. Results

### 3.1 Strength characteristics

Brazilian splitting tests were carried out on single rock specimens and single concrete specimens in order to complete a comparative analysis. Fig 3 shows the stress-strain curves of the single rock specimens and single concrete specimens under the Brazilian splitting tests. It can be seen from the diagram that the splitting tensile strength levels of the concrete samples were greater than those of the rock samples. The maximum splitting strength of the rock samples in three groups of independent repeated tests was 2.32 MPa and the minimum was 2.21 MPa, with an average splitting strength was 2.26 MPa. The maximum splitting strength of the concrete specimens was 3.26 MPa, and the minimum was 2.74 MPa, with an average splitting strength of 3.04 MPa.

According to the value of the splitting tensile strength of each group of roughness, the median sample is selected. Fig 4(A) shows the stress-strain curve of the rock-concrete specimen splitting process in the roughness range of $f_0$ to $f_5$. It can be seen that the stress-strain curve of rock-concrete specimen under radial concentrated load can be roughly divided into three stages: compaction stage, elasticity stage and failure stage.

Fig 4(B) shows the relationship between splitting peak tensile strain, compressive strain and roughness of rock-concrete specimens. The interface roughness mode controls the upper and lower limits of tensile strain. The six roughness are divided into two groups according to whether there are serrations in the center. When the roughness is $f_0, f_2, f_4$, there are no serrations in the center of the disc. When the roughness is $f_1, f_3, f_5$, there are serrations in the center

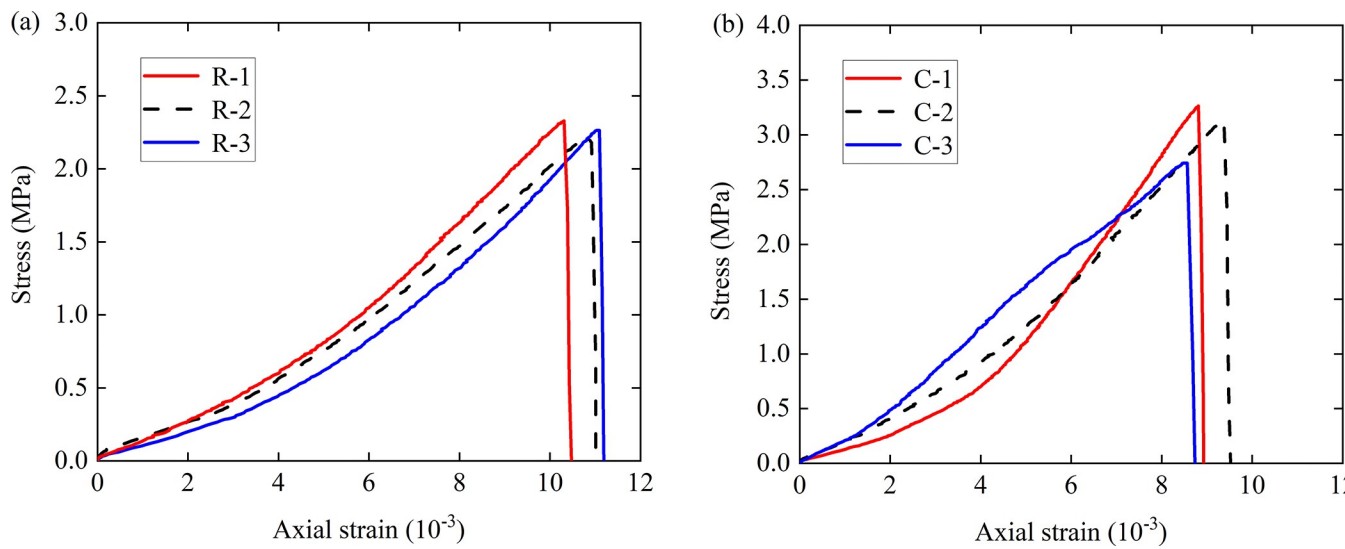

**Fig 3.** The whole stress-strain curves of different kinds of samples, (a) rock samples; (b) concrete samples.

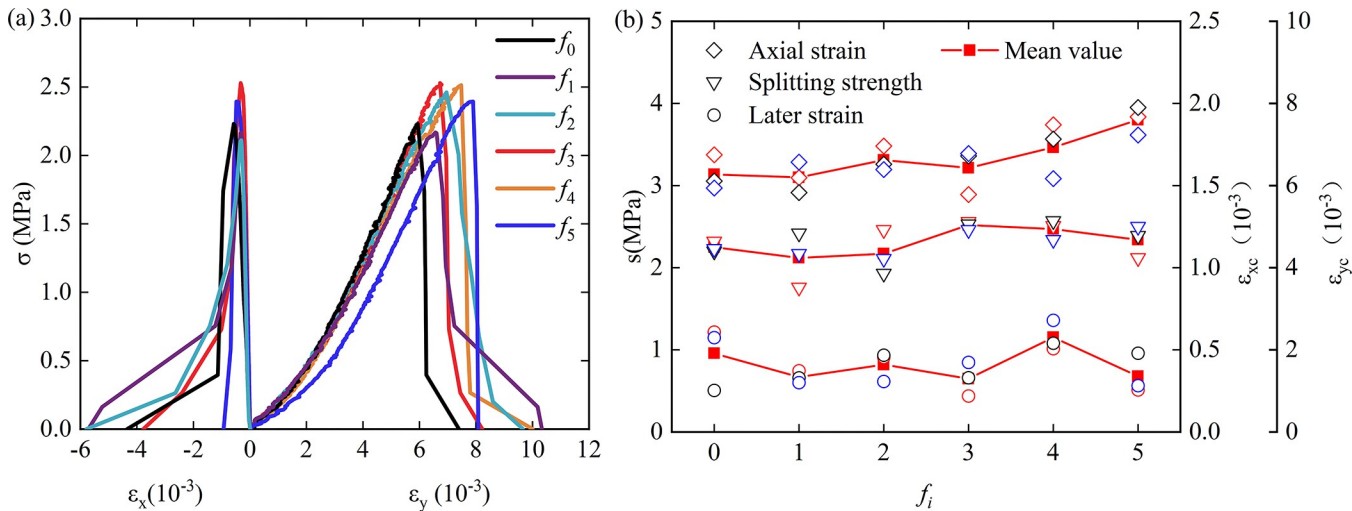

**Fig 4.** Stress-strain curves of rock-concrete specimens with different roughness, (a) Stress-strain curve; (b) Peak stress and strain red symbol is $f_i$-1, black symbol is $f_i$-2, blue symbol is $f_i$-3(the same below).

of the disc. When there are serrations in the center, the peak value of tensile strain is often smaller than that without serrations, indicating that there are serrations in the center, and the constraint ability to time lateral deformation is enhanced. The reason may have a great relationship with the stress condition in the center of the sample. It is known that the tensile strength of concrete, the tensile strength of rock and the tensile strength of rock-concrete interface decrease in turn. When there is no sawtooth in the center of the sample, the most central position of the sample is the rock-concrete interface, and the binding force is small. When there is sawtooth in the center of the sample, the most central position of the sample is concrete, and the binding force is large. Similarly, there is a horizontal interface between the concrete sawtooth and the sandstone in the center of the sample. When subjected to tension, it can provide a certain horizontal friction force, which greatly increases the binding force of the horizontal deformation of the sample. The two sets of different levels of data alternate with each other to form a letter W-shaped trend. As the roughness increases, the compressive strain tends to increase as a whole, but when the roughness is $f_2$, his average value is higher than the data on both sides, so that the overall approximate smooth rise curve has a raised small peak. The sample and the non-aliasing group have a more obvious and smoother increase trend than the whole.

### 3.2 Splitting energy rate

The accumulation and release of energy runs through the whole process of rock deformation and failure. In the process of gradual action of external force, the rock is gradually damaged, and the strength is weakened and finally lost. Studying the law of energy change inside the rock can reflect the degree of attenuation of the original strength of the material and reveal the development law of cracks inside the rock [22]. The loading process of the rock-concrete sample is in a complex stress-strain environment. The combined sample first accumulates energy continuously, and the original mechanical equilibrium state is destroyed after reaching the peak value. The rock mass produces new deformation, even fracture and fragmentation, and the rock mass releases energy during the stress unloading process.

The loading process of a composite specimen is the continuous operation process of the testing machine. The composite specimen will continuously accumulate energy, which can be

determined by the area enclosed by the load-displacement curve and the abscissa as follows:

$$W_i = \int_0^u p_i dy_i \tag{2}$$

Where $p_i$ is the corresponding load (N) at the sampling point, $y_i$ is the compression deformation (mm); $i$ is the sampling point; and $u$ is the maximum deformation (mm).

The actual calculation shown in Eq (2) was based on the definite integral method, and the load-displacement curve of the test sample is calculated by the area of a small trapezoidal strip. Due to the differences in the thicknesses of the disc samples tested in this study, the energy accumulated per unit rupture area (also known as the energy rate W/ (J·m$^{-3}$)) was calculated as follows:

$$W = \frac{1}{2DH} \sum_{i=1}^{n} (P_{i+1} + P)_i (y_{i+1} - y_i) \tag{3}$$

Where $n$ is the number of test samples; $P_{i+1}$ is the corresponding load at the adjacent point of the sampling point (N); and $y_{i+1}$ is the compression deformation (mm).

Fig 5 shows the relationship between the accumulated energy before the peak and the compressive strain during the splitting process of rock-concrete specimens with different roughness. The selected sample is consistent with the sample in Fig 3(A). It can be seen from the diagram that during the loading process of the rock-concrete sample, the pre-peak energy storage rate of the rock-concrete sample increases continuously with the increase of the compressive strain. In the early stage of the test, the speed of increasing strain is faster and the speed of increasing stress is smaller in the compaction stage. In the elastic stage, the slope of the stress-strain curve is stable, so the curve of the relationship between the pre-peak energy storage and the strain shows a downward trend. As the load continues to increase, the pre-peak energy storage of the rock-concrete sample is roughly linear with the strain, and the energy storage increases rapidly. When the slope of $f_0$ to $f_4$ curve is approximate, the peak compressive strain determines the amount of energy accumulated. The energy curve in the case of $f_5$ grows slowly compared with other curves, and is always lower than other curves under the same compressive strain condition, indicating that the ability to accumulate energy is greatly weakened.

It can be seen from the average value of the pre-peak accumulated energy in Fig 6(A) that when the roughness decreases from $f_0$ to $f_1$, it increases from $f_1$ to $f_4$, and the rising trend is accelerated. When the roughness reaches $f_3$, the average value exceeds $f_0$, and decreases sharply from $f_4$ to $f_5$, indicating that the stress change leads to a rapid decline in the ability to accumulate energy when the interface changes from no sawtooth to sawtooth. Then, with the increase of roughness, the accumulated energy increases rapidly. After reaching the highest point, the material limit is reached, and the increase of roughness causes a reaction instead. Similarly, the change trend of post-peak release energy is opposite to that of pre-peak accumulated energy.

It can be seen from Fig 6(B) that when the roughness is from $f_0$ to $f_1$, the value drops sharply. Compared with the pre-peak accumulation energy, $f_1$ to $f_5$ shows an upward trend, and the rate of increase of $f_1$ to $f_3$ is increasing. From $f_3$ to $f_4$, the rate of increase suddenly slows down until $f_5$, showing a trend of increasing more and more slowly. The dissipated energy is the difference between the accumulated energy before the peak and the elastic energy, which reflects the internal damage of the sample to a certain extent. The dissipated energy becomes larger when the roughness is from $f_0$ to $f_1$, and the dissipated energy becomes smaller and reaches the lowest when $f_1$ to $f_3$. During the $f_3$ to $f_5$ period, due to the relatively gentle increase of elastic energy, the change trend of dissipated energy at this stage is consistent with the accumulated energy before the peak, both of which increase first and then decrease. Under

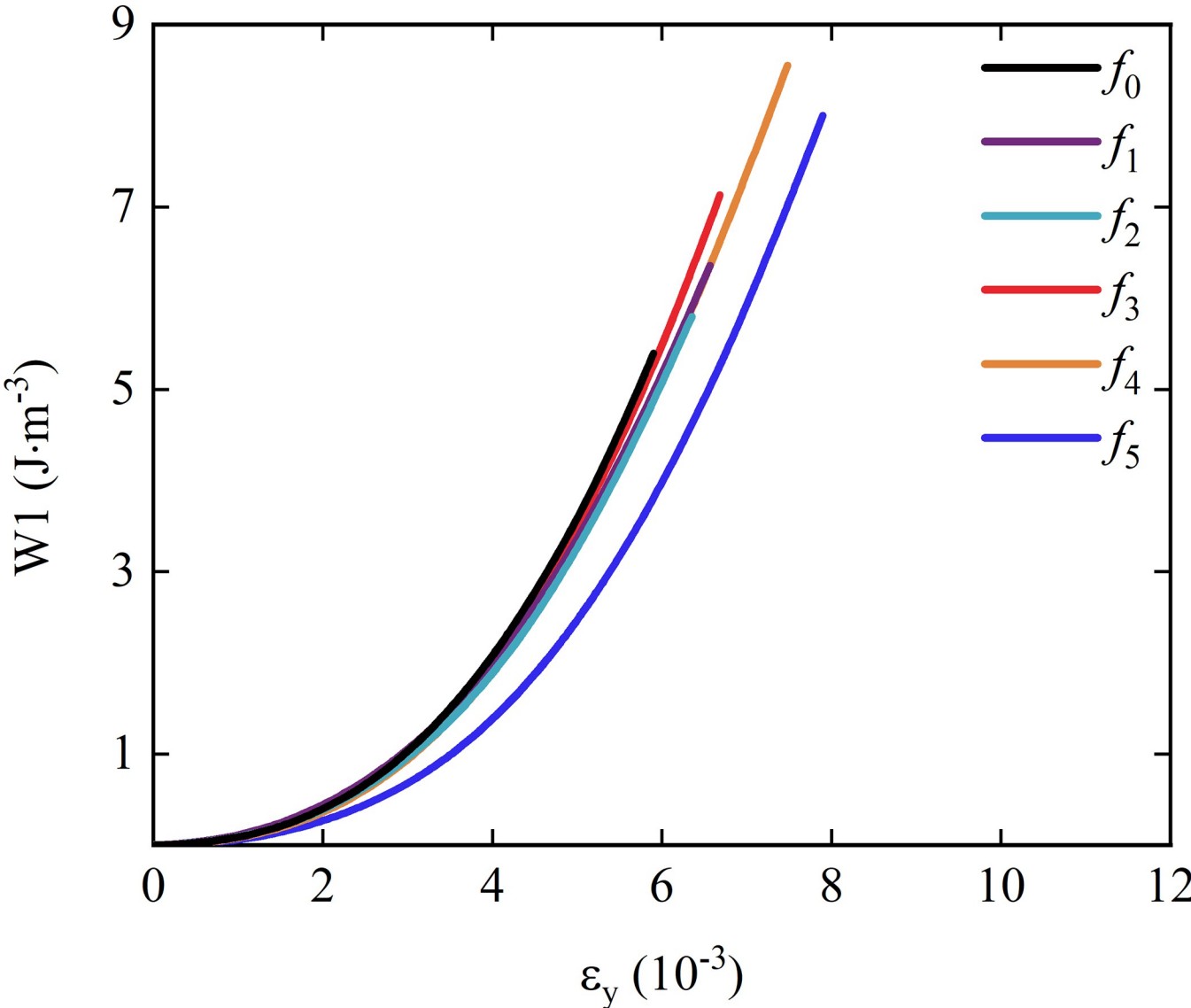

**Fig 5. The relationship between pre-peak accumulated energy and compressive strain of rock-concrete specimens with different roughness.**

the influence of the increase of elastic energy, the change of dissipated energy is more gentle than that of accumulated energy before the peak.

## 3.3 Crack evolution characteristics

The failure processes of the rock materials undergoing external force are often accompanied by the expansion of internal cracks in the rock. In order to better understand the development and change law of fractures in rock materials during loading, Martin introduced a concept of crack strain. It was determined that the crack strain values could be utilized to determine the fracture degrees of rock masses during loading, thereby enabling the study of the progressive damage behaviors of rock masses. Fracture stress is generally used to quantitatively analyze the degree and number of cracks in rock masses. Fracture stress refers to the axial and circumferential deformations caused by the initiation, extension, and penetration of primary cracks, as

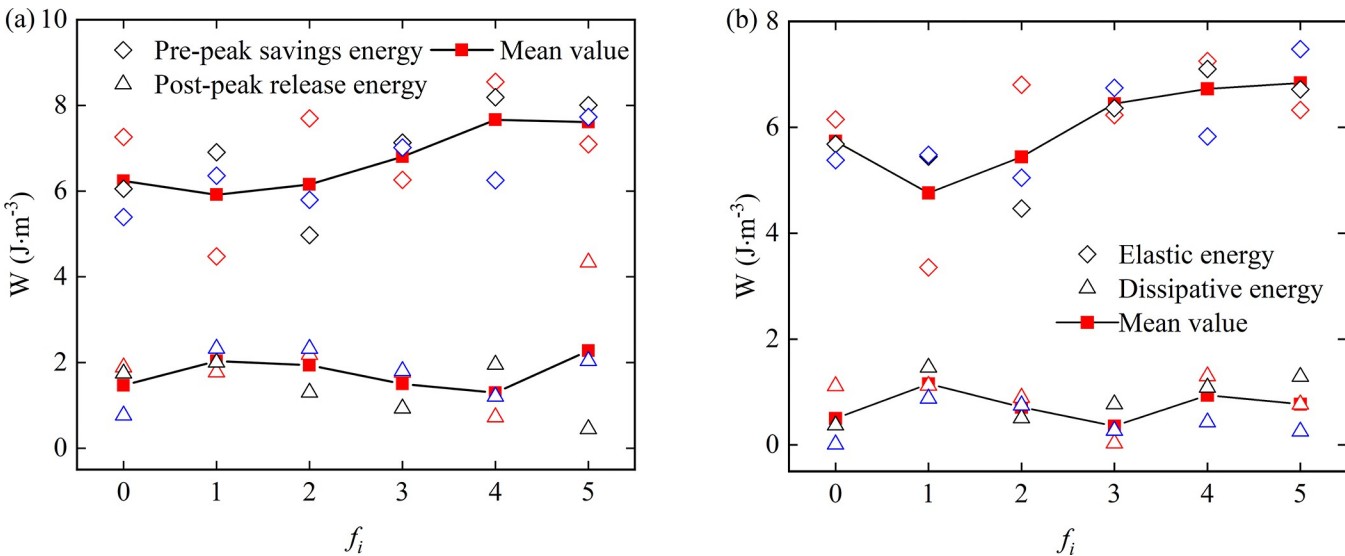

**Fig 6.** The relationship between various energy and roughness in the splitting process of rock-concrete specimens, (a) Pre-peak savings energy, released energy after the peak; (b) Elastic energy, dissipated energy.

well as the initiation of new cracks under the influence of external pressure [23]. The calculation formula of crack axial strain is as follows:

$$\varepsilon_1^c = \varepsilon_1 - \frac{\sigma_1}{E} \tag{4}$$

where $\varepsilon_1^c$ represents the crack axial strain, $\varepsilon_1$ is the axial strain, $\sigma_1$ denotes the axial stress, and $E$ is the elastic modulus.

Fig 7(A) shows the relationship between crack parameters and roughness under the splitting test of rock-concrete specimens. The selected sample is consistent with the sample in Fig 2 (A).

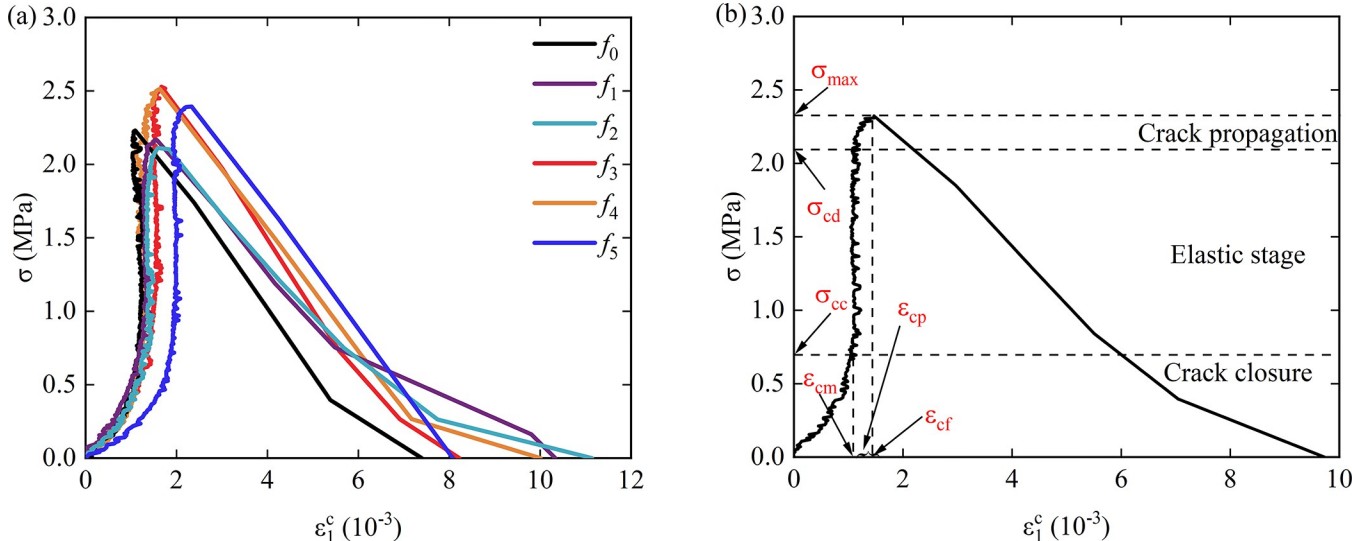

**Fig 7.** Stress-crack axial strain curve, (a) Stress-crack axial strain curve with different roughness; (b) Crack parameter calculation schematic diagram.

Taking sandstone sample $f_0$-1 as an example, Fig 2(B) shows the calculation diagram of axial crack parameters, where $\sigma_{cc}$ is the axial crack closure stress, $\sigma_{cd}$ is the axial crack damage stress, $\varepsilon_{cm}$ is the axial crack closure strain, $\varepsilon_{cf}$ is the axial crack peak strain. In order to study the law of crack propagation in sandstone during uniaxial compression, the difference between axial crack peak strain and axial crack closure strain was calculated, and the axial crack propagation strain $\varepsilon_{cp}$ was obtained as follows:

$$\varepsilon_{cp} = \varepsilon_{cf} - \varepsilon_{cm} \tag{5}$$

According to the stress-crack strain curve in Fig 7(B), it can be seen that the pre-peak curve is divided into three stages: crack closure stage, elastic stage and crack propagation stage. The elastic stage curve is almost perpendicular to the abscissa, indicating that the crack develops rapidly, the axial strain of the crack increases slightly, and the stress increases rapidly.

It can be seen from Fig 8 that the average values of crack closure strain, crack peak strain, crack propagation strain and crack damage stress have a sinusoidal periodic variation relationship with roughness. In Fig 8(A) and 8(B), $f_0$ begins to rise and reaches the peak between $f_1$ and $f_2$, then decreases to the valley between $f_3$ and $f_4$, and then rises to the second peak near $f_5$. Fig 8(C) reaches the first peak near $f_1$, drops to the valley near $f_3$, and reaches the second peak between $f_4$ and $f_5$. The crack damage strain in Fig 8(D) shows the opposite trend to the previous three figures, and the crack damage strain tends to decrease when the crack strain increases. The crack closure strain increases first and then decreases slowly with the increase of roughness.

## 4. Discussion

In practical engineering applications, the roughness of the rock surface is increased by groove treatment on the rock surface to improve the bonding performance of the rock-concrete interface. Since concrete and rock are two different materials, the stress state at the interface is more complicated than that of a single material. If the most suitable number of grooves can be designed according to the mechanical properties of rock or concrete, the safety and durability of the structure can be effectively guaranteed.

In order to obtain the most suitable interface roughness on the rock, six kinds of rock surface groove number are designed and the roughness of the corresponding groove number is calculated. It was found that with the increase of roughness, the proportion of concrete in the cross section of the specimen increased significantly. It shows that with the increase of roughness, the stress concentration area tends to shift from the rock area to the concrete area, and the concrete begins to replace the rock to destroy, which plays a role in protecting the rock. The interface between the serrations receives tangential friction during tension.

The typical Brazilian splitting failure mode is shown in Fig 9. There is tensile failure along most of the diameter length range, and a small range of shear failure may occur in a particularly small range at both ends of the loading.

Fig 10 shows the splitting failure mode of the disc specimen under different roughness. Under the condition of roughness $f_0$, because the tensile strength of rock material is lower than that of concrete material, near the center of rock-concrete interface, when the concrete material reaches the ultimate bearing capacity earlier than the rock-concrete cementation surface, the rock near the interface first undergoes tensile failure, and the crack develops in the direction of compression, and the stress concentration at both ends of compression causes shear failure ($f_0$-1,$f_0$-2); in this process, when the bonding surface reaches the ultimate bearing capacity, the crack develops along the interface and the tensile failure of the interface occurs ($f_0$-3).

When the roughness is from $f_0$ to $f_1$, when a notch is added to the interface, the stress state at the original interface is changed compared with the condition of $f_0$, and the stress

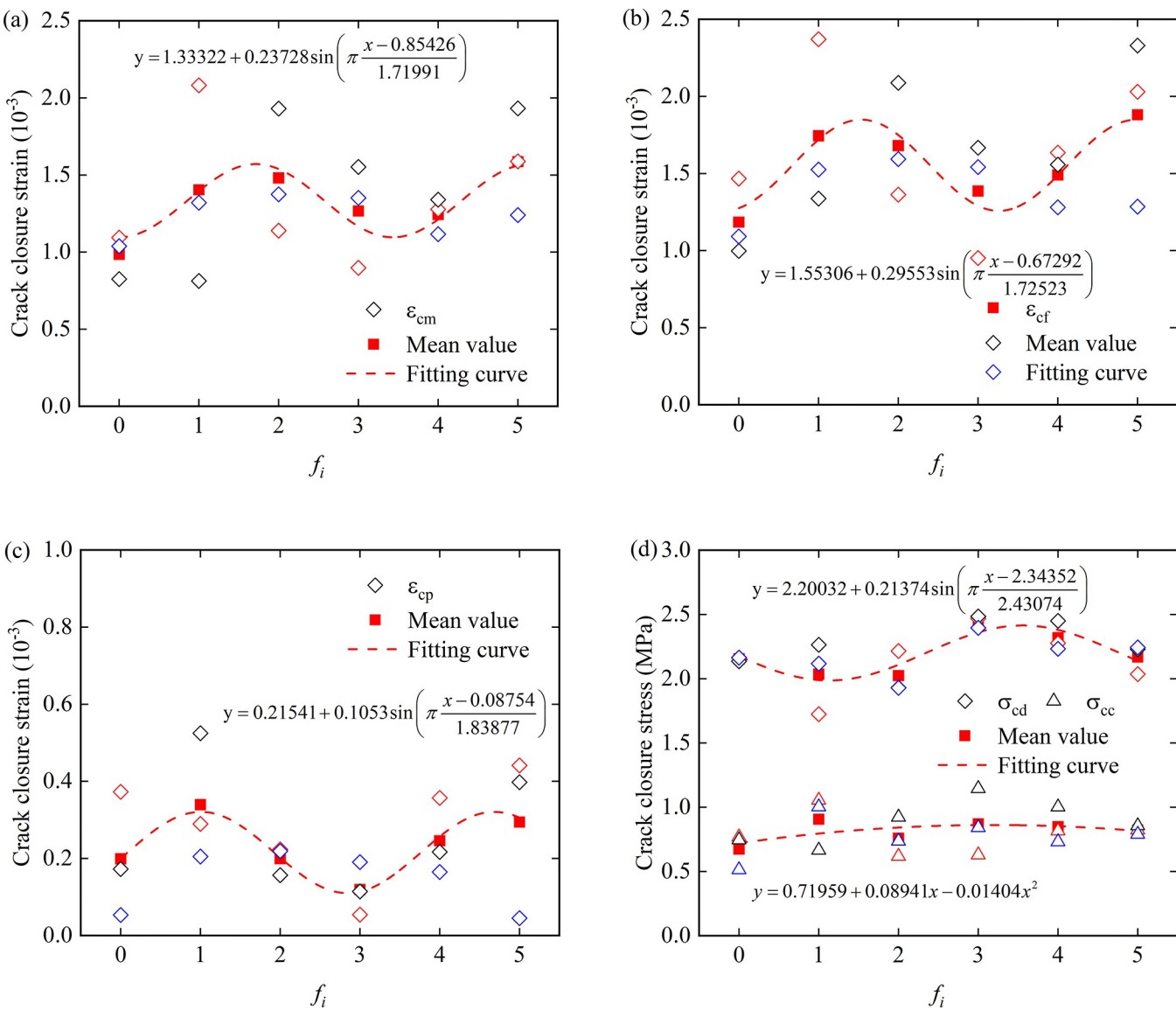

**Fig 8.** Relationship between roughness and rock-concrete crack parameters, (a) crack closure strain; (b) crack peak strain; (c) crack propagation strain; (d) crack closure stress and crack damage stress.

concentration is generated near the notch. The central crack area starts to crack near the notch and develops to both ends of the pattern under compression. At this time, the strength of the concrete serration reaches the ultimate bearing capacity of the concrete serration, and the concrete serration breaks. The aggregate distribution in the serrated area affects the strength of the serrated, and a good aggregate particle size distribution can enhance the overall strength of the concrete serrated. If the aggregate particle size is too large or the needle-like aggregate is mixed, the stress concentration inside the serrated may be caused and the serrated strength is reduced.

When the roughness is from $f_2$ to $f_3$, the average splitting strength reaches the highest, and the proportion of concrete in the fracture surface of $f_3$-1 and $f_3$-3 begins to increase significantly, and the cross section of $f_3$-1 can be seen that the crack appears throughout the concrete.

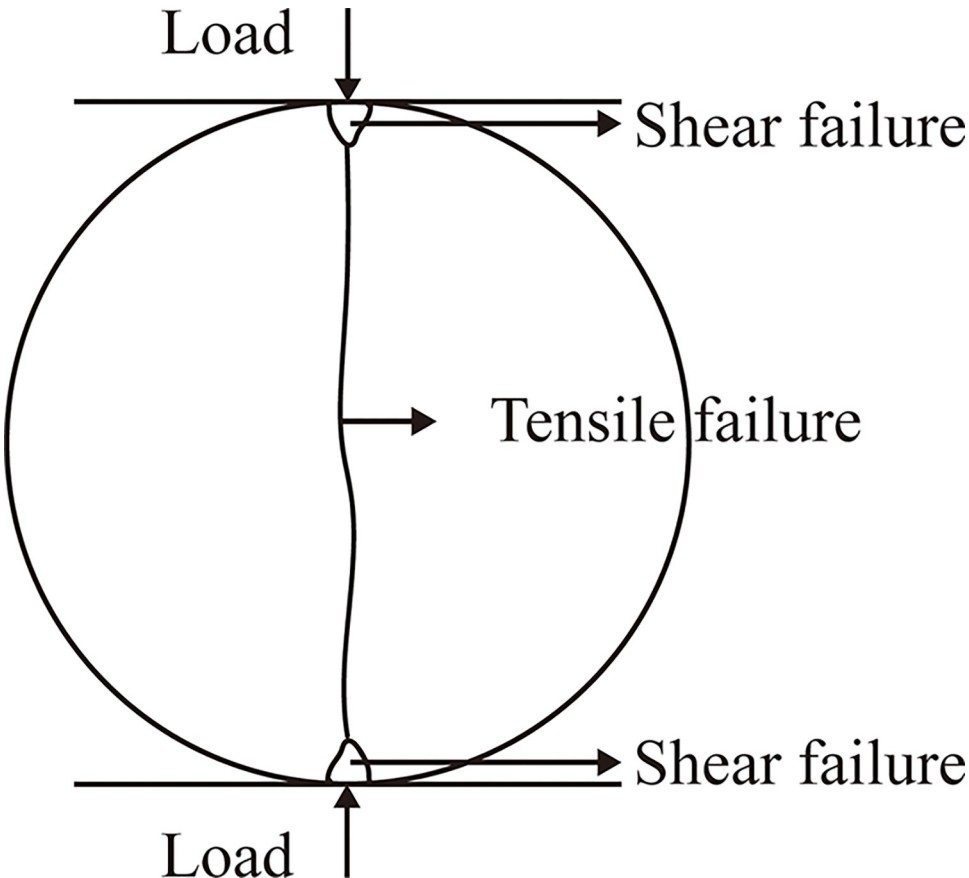

**Fig 9. The typical Brazilian splitting failure mode [24].**

Two cracks are derived near the central serration of the $f_3$-2 specimen, one of which extends in an approximate vertical loading direction and develops along the interface when it develops to the interface, and the other develops interrupted when it extends to another serration, which is similar to the development trend of the left crack. It shows that the rock-concrete interface is more prone to tensile failure, and it also shows that the method of grooving can effectively slow down the development of cracks and enhance the tensile strength of rock-concrete.

When the roughness is from $f_3$ to $f_4$, the crack of $f_4$-2 penetrates the concrete and forms a crack approximately horizontal to the loading direction. The crack of $f_4$-3 penetrates the rock and forms an arc crack. The crack of $f_4$-1 develops from the center to both ends. Half of the crack is generated on the interface and the other part is generated in the rock.

When the roughness is from $f_4$ to $f_5$, the proportion of concrete on the crack fracture surface increases, the $f_5$-1 crack breaks along the interface, and the serrated fracture occurs. The $f_5$-2 crack basically penetrates through the concrete area to form an arc crack.

## 5. Conclusion

Based on the theory of rock tensile strength, this paper realizes comprehensive analysis. During the test, the roughness of the rock-coagulation interface is changed, and its influence on the tensile strength of the rock-concrete interface is discussed. The following conclusions can be drawn:

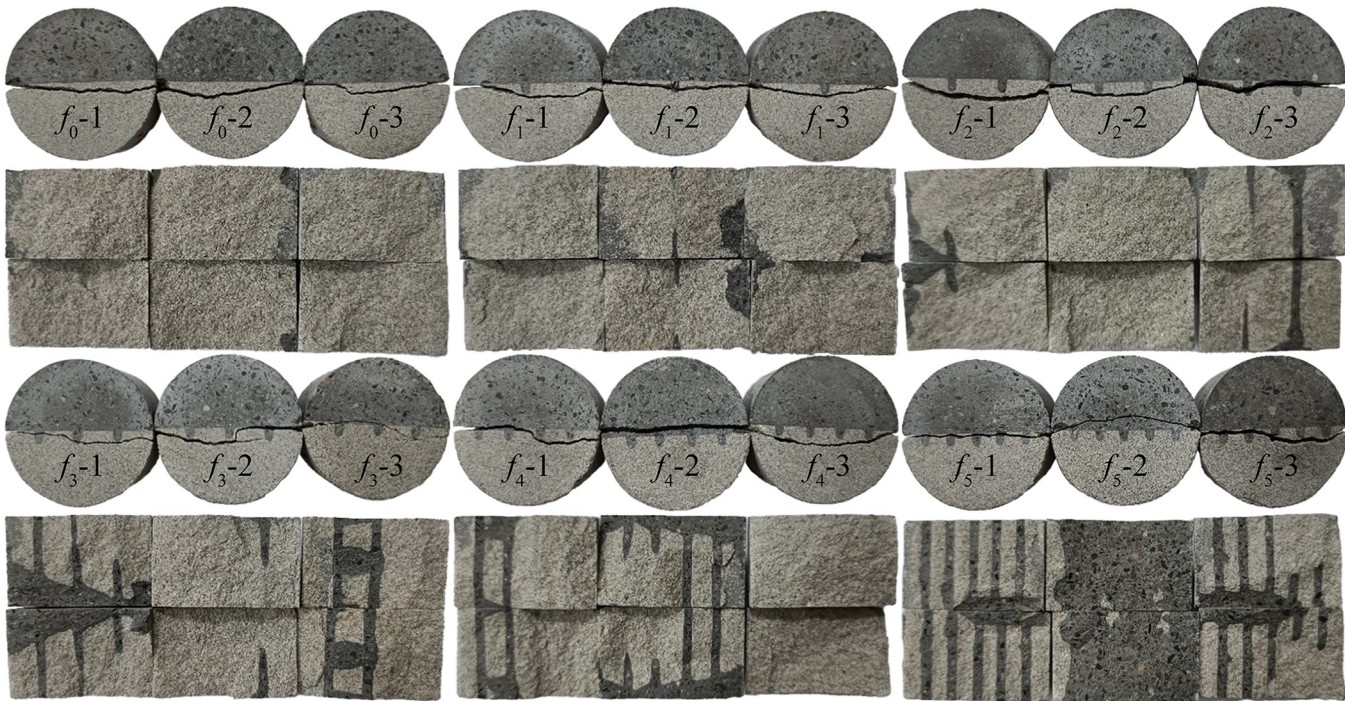

**Fig 10. Splitting failure mode of rock-concrete specimen with different roughness.**

1. The roughness interface mode controls the transverse strain of the sample. When there are serrations in the center of the sample, the transverse strain is constrained to increase. The roughness size controls the axial strain of the sample, and the axial strain increases with the increase of roughness. The roughness size and the interface mode influence each other. The roughness size controls the overall trend of the strain, and the roughness interface mode determines the ability to limit the strain.

2. In the range of $f_1$ to $f_4$, the pre-peak cumulative energy of rock-concrete composite increases gradually with the increase of roughness, and the post-peak release energy decreases with the increase of roughness. In the range of $f_1$ to $f_5$, the elastic energy increases with the increase of roughness, and the increase rate increases first and then decreases. The change of dissipation energy is highly positively correlated with the change of crack propagation strain.

3. The crack closure strain, crack peak strain, crack propagation strain and crack damage stress have a sinusoidal periodic variation relationship with roughness. The first three peak-valley regions are approximately negatively correlated with the latter, and the crack damage stress has a high correlation with the change of the splitting strength of the sample.

4. The size of the roughness and the roughness mode should be carefully selected. If the increased roughness is too small or too large, the increase of cracks will lead to the decrease of strength. If the roughness is too large, it will increase the workload and cause too much damage to the original rock. The deformation of the sample can be well constrained by groove treatment at the part of the sample with large stress.

5. The method of increasing the roughness of the rock-concrete interface by grooving. As the roughness increases, the proportion of concrete damage increases. To a certain extent, it

reflects that the stress concentration area transitions from the rock area to the concrete area as the roughness increases.

## Supporting information

**S1 Data.**
(XLSX)

## Author Contributions

**Data curation:** Jiahao Wang.

**Formal analysis:** Jiahao Wang.

**Funding acquisition:** Yan Chen.

**Investigation:** Gaofei Wang.

**Methodology:** Yan Chen.

**Project administration:** Yan Chen.

**Software:** Lei Zhou.

**Validation:** Liangtao Deng.

**Visualization:** Gaofei Wang, Lei Zhou.

**Writing – original draft:** Yan Chen, Gaofei Wang.

**Writing – review & editing:** Yan Chen.

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
