## [Decision Letter · Decision Letter 0]

3 Jan 2024

PONE-D-23-40490Effects of different roughness on brazilian splitting characteristics of rock-concrete interfacePLOS ONE

Dear Dr. Chen,

Thank you for submitting your manuscript to PLOS ONE. After careful consideration, we feel that it has merit but does not fully meet PLOS ONE’s publication criteria as it currently stands. Therefore, we invite you to submit a revised version of the manuscript that addresses the points raised during the review process.

Please address all the comments one by one. 

We look forward to receiving your revised manuscript.

Kind regards,

Xun Xi

Academic Editor

PLOS ONE

Journal Requirements:

4. Thank you for stating the following in the Acknowledgments Section of your manuscript: "Financial supports from the National Natural Science Foundation of China (51904092), Key Scientific Research Project of Higher Education Institutions in Henan Province (Grant No. 24A440004), the Fundamental Research Funds for the Universities of Henan Province (NSFRF230403, NSFRF210454), Young backbone teachers funding program of Henan Polytechnic University (2022XQG-01), the research fund of Henan Key Laboratory for Green and Efficient Mining & Comprehensive Utilization of Mineral Resources (KCF2202) and the research fund of Jiaozuo Road Traffic and Transportation Science and Technology research center (JRTT2023004, JRTT2023010, JRTT2023011) are gratefully acknowledged."

Please remove any funding-related text from the manuscript and let us know how you would like to update your Funding Statement. Currently, your Funding Statement reads as follows: "The authors received no specific funding for this work."

5. We note that your Data Availability Statement is currently as follows: "ALL relevant data are within the manuscript and its Supporting Information".

6. Please amend the manuscript submission data (via Edit Submission) to include authors Lei Zhou, Liangtao Deng, and Jiahao Wang.

Reviewers' comments:

Reviewer's Responses to Questions

**Comments to the Author**

1. Is the manuscript technically sound, and do the data support the conclusions?

Reviewer #1: Yes

Reviewer #2: Partly

2. Has the statistical analysis been performed appropriately and rigorously? 

Reviewer #1: Yes

Reviewer #2: Yes

3. Have the authors made all data underlying the findings in their manuscript fully available?

Reviewer #1: Yes

Reviewer #2: Yes

4. Is the manuscript presented in an intelligible fashion and written in standard English?

Reviewer #1: Yes

Reviewer #2: Yes

5. Review Comments to the Author

Reviewer #1: Based on the Brazilian splitting experiment, this paper studies the effects of six different roughness on rock-concrete properties, analyzes the influence of interface roughness on the tensile strength, deformation, splitting energy rate and failure mode of rock-concrete composites. Although the test is simple, the analysis is acceptable. However, the manuscript has some problems and it would require some major revisions. The specific issues are as follows:

1、The writing of the “1 Introduction” lacks the reference of roughness to the characteristics of coal-rock combination and suggests the author consult relevant literature and supplement them in time.

2、“roughness” is an uncountable noun.

3、“The peak tensile strain is often smaller than that without serrations, indicating that the center has serrations” the relationship between “peek tensile strain” and “serrations” need to be further explanation.

4、I suggest the author should increase the basic mechanical parameters of rock and concrete and increase the integrity of the paper.

5、The author should be given the calculation formula of crack evolution parameters.

6、I suggest the author give the fitting formula of Fig.6.

Reviewer #2: I suggest to the Editor that the manuscript requests a “major revision.” The detailed review comments for improving the manuscript have been uploaded as an attachment. Please refer to the attachment to make revisions.

6. PLOS authors have the option to publish the peer review history of their article (what does this mean?). If published, this will include your full peer review and any attached files.

Reviewer #1: No

Reviewer #2: No

---

## [Author Response · Author response to Decision Letter 0]

29 May 2024

Reviewer 1

Based on the Brazilian splitting experiment, this paper studies the effects of six different roughness on rock-concrete properties, analyzes the influence of interface roughness on the tensile strength, deformation, splitting energy rate and failure mode of rock-concrete composites. Although the test is simple, the analysis is acceptable. However, the manuscript has some problems and it would require some major revisions.

Answer:

Thanks very much for your kind suggestion. We deeply appreciate the comments concerning the manuscript. Revisions in the text are shown using red highlight for additions, and blue highlight for grammatical checks.

Comment 1: The writing of the “1 Introduction” lacks the reference of roughness to the characteristics of coal-rock combination and suggests the author consult relevant literature and supplement them in time.

Answer: Revision at Page 2 Line 43~45.

Thanks very much for your kind suggestion. We have rewritten the abstract in the revised manuscript, giving a concise description of the paper and listing some new findings.

Comment 2: Please introduction the significance of the effect of rock-concrete failure to the tunnel engineering.

Answer: Revision at Page 2 Line 33~46; Page 3 Line 47~48.

Thank you for the reviewer’s comment. We have introduced the influence of rock-concrete failure to tunnel engineering in the introduction of the revised manuscript.

Comment 3: “The peak tensile strain is often smaller than that without serrations, indicating that the center has serrations” the relationship between “peek tensile strain” and “serrations” need to be further explanation.

Answer: Revision at Page 11 Line 207 ~ 210; Page 12 Line 211 ~ 219.

Thank you very much for your friendly advice. We have supplemented the relationship between “peek tensile strain” and “serrations” in the revised draft, and explained the reasons for this phenomenon.

Comment 4: I suggest the author should increase the basic mechanical parameters of rock and concrete and increase the integrity of the paper.

Answer: Revision at Page 10 Line 185~192; Page 11 Line 193~196.

We really appreciate the reviewer’s suggestion. We supplemented the Brazilian splitting data of sandstone and concrete in the revised manuscript.

Comment 5: The author should be given the calculation formula of crack evolution parameters.

Answer: Revision at Page 16 Line 303~305; Page 17 Line 306~327; Page 18 Line 328.

Thanks to the reviewer’s suggestion. We supplemented the calculation formula of crack propagation parameters in the revised manuscript.

Comment 6: I suggest the author give the fitting formula of Fig.6.

Answer: Revision at Page 19 Line 346~347.

Thank you for the reviewer’s advice. We supplemented the fitting formula of Fig.6 in the revised manuscript.

Reviewer 2

This paper (PONE-D-23-40490) explores the influence of roughness on the tensile 

properties of rock-concrete composite disc specimens. The sinusoidal periodic variation 

of crack-related parameters with roughness is an interesting observation. This paper 

offers valuable insights for practical applications, especially in construction projects 

involving rock-concrete interfaces. However, the experimental methods used in this 

study were simplistic and lacked depth of research. I suggest to the Editor that the 

manuscript requests a “major revision.” Some specific comments for improving the 

manuscript are given as follows.

Answer:

Thanks very much for your kind suggestion. We have revised the manuscript according to each review comment. Revisions in the text are shown using red highlight for additions, and blue highlight for grammatical checks.

Comment 1: Introduction: The origin and significance of the Brazilian Splitting Test in the field of rock mechanics are considered common knowledge, requiring minimal elaboration. Considering this, the author is encouraged to supplement the introduction with content directly pertinent to the research focus of this manuscript. By providing additional context on the specific relevance of the research content, such as its implications in practical engineering applications, the introduction can be enriched to better engage the reader and underscore the importance of the investigation.

Answer: Revision at Page 2 Line 33~46; Page 3 Line 47~68; Page 4 Line 69~77.

Thanks very much for your kind suggestion. We have rewritten a new introduction, deleted the origin and significance of the Brazilian splitting test in the field of rock mechanics, and supplemented the relevant content related to the focus of this study.

Comment 2: As the author acknowledges in the Introduction, numerous scholars have employed advanced testing techniques, such as AE, DIC, and transparent materials, to observe the initiation and propagation of cracks. In contrast, this article does not leverage these sophisticated testing methods for supplementary investigation. Therefore, in comparison to existing research, what constitutes the innovative aspects of this article?

Answer:

Thanks to the reviewer’s suggestion. Unfortunately, our laboratory lacks precision instruments such as AE, DIC, and transparent materials. Therefore, in our tests, we cannot monitor the evolution of failure. In future research, we will seek the help of others to obtain more advanced equipment to monitor the failure process of the specimen and to analyze the failure characteristics of the specimen more accurately. The innovation of this paper is to use the method of artificial groove to make rock-concrete composite disc samples with different roughness, and to quantify the roughness value of the samples under different working conditions by sand filling method, so as to study the influence of roughness on the tensile properties of rock-concrete composite disc samples.

Comment 3: Sample making: Provide additional details on the specific methods employed for artificially grooving the specimens. This could include information on the tools, techniques, and parameters used in the grooving process. Suggest adding images or schematics that can illustrate the groove treatment process on the rock-concrete specimens. This can provide readers with a clearer understanding of the 

experimental setup.

Answer: Revision at Page 5 Line 110~111; Page 6 Line 112~122; Page 8 Line 157~159; Page 9 Line 160; Page 9 Line 171~179.

Thank you for the reviewer’s advice. We do not have a detailed description of the specific methods of manual grooving samples. After verification with the sample processing party, we give a more detailed description of the tools, parameters and related technologies used in the grooving process. In the revised manuscript, we supplemented the process flow diagram of the grooving process and the related schematic diagrams of the grooving samples with different roughness, hoping to provide readers with a more concise test introduction and facilitate readers to quickly and clearly understand the experimental setup.

Comment 4: Splitting energy rate: The term "energy" mentioned by the author refers to what exactly? If it pertains to elastic strain energy, is the strain limited to axial strain εy alone, or does it also encompass horizontal strain εx? Kindly provide a more elaborate explanation of the concept of energy and the specific calculation methods.

Answer: Revision at Page 16 Line 303~305; Page 17 Line 306~327; Page 18 Line 328.

Thanks very much for your kind suggestion. We explain the energy in the newly revised manuscript, and explain the related concepts and specific calculation methods of energy in detail. The elastic strain energy designed and calculated in this paper is only related to the axial strain εy.

Comment 5: Crack evolution characteristics: The author's classification of the crack propagation process solely based on the stress-strain curve appears overly simplistic. If experimental conditions permit, it is recommended to consider employing techniques such as AE or DIC for supplementary investigation. This would add depth to the research results, enhancing both the reliability and comprehensiveness of the study.

Answer: Thanks to the reviewer’s suggestion. As mentioned in Answer 2, our laboratory lacks precision instruments such as AE, DIC, and transparent materials, so the content we discuss in this paper mainly focuses on the stress-strain curve of the sample. In future experiments, our research group will consider using more test instruments and better test conditions to do deeper research to improve the reliability and comprehensiveness of the research.

---

## [Decision Letter · Decision Letter 1]

2 Jul 2024

Effects of different roughness on brazilian splitting characteristics of rock-concrete interface

PONE-D-23-40490R1

Dear Dr. Chen,

We’re pleased to inform you that your manuscript has been judged scientifically suitable for publication and will be formally accepted for publication once it meets all outstanding technical requirements.

Kind regards,

Xun Xi

Academic Editor

PLOS ONE

Additional Editor Comments (optional):

Reviewers' comments:

Reviewer's Responses to Questions

**Comments to the Author**

1. If the authors have adequately addressed your comments raised in a previous round of review and you feel that this manuscript is now acceptable for publication, you may indicate that here to bypass the “Comments to the Author” section, enter your conflict of interest statement in the “Confidential to Editor” section, and submit your "Accept" recommendation.

Reviewer #1: All comments have been addressed

Reviewer #2: All comments have been addressed

2. Is the manuscript technically sound, and do the data support the conclusions?

Reviewer #1: Yes

Reviewer #2: Yes

3. Has the statistical analysis been performed appropriately and rigorously? 

Reviewer #1: Yes

Reviewer #2: Yes

4. Have the authors made all data underlying the findings in their manuscript fully available?

Reviewer #1: Yes

Reviewer #2: Yes

5. Is the manuscript presented in an intelligible fashion and written in standard English?

Reviewer #1: Yes

Reviewer #2: Yes

6. Review Comments to the Author

Reviewer #1: The review comments have been responded to and revised, and this article can be considered for acceptance.

Reviewer #2: The revised version of this article has addressed the concerns raised in the previous review. The authors have meticulously incorporated the suggestions, resulting in a significantly improved manuscript. The manuscript is now suitable for publication.

7. PLOS authors have the option to publish the peer review history of their article (what does this mean?). If published, this will include your full peer review and any attached files.

Reviewer #1: No

Reviewer #2: No

---

## [Editor Report · Acceptance letter]

10 Jul 2024

PONE-D-23-40490R1 

PLOS ONE

Dear Dr. Chen, 

I'm pleased to inform you that your manuscript has been deemed suitable for publication in PLOS ONE. Congratulations! Your manuscript is now being handed over to our production team.

Kind regards, 

on behalf of

Dr. Xun Xi 

Academic Editor

PLOS ONE